# ORGAN-DETR: 3D ORGAN DETECTION TRANSFOMER WITH MULTISCALE ATTENTION AND DENSE QUERY MATCHING

## ABSTRACT

Query-based Transformers have been yielding impressive results in object detection. The potential of DETR-like methods for 3D data, especially in volumetric medical imaging, remains largely unexplored. This study presents Organ-DETR that contains two novel modules, MultiScale Attention (MSA) and Dense Query Matching (DQM), for boosting the performance of DEtection TRansformers (DETRs) for 3D organ detection. MSA introduces a novel top-down representation learning approach for efficient encoding of 3D visual data. MSA has a multiscale attention architecture that leverages dual self-attention and cross-attention mechanisms to provide the most relevant features for DETRs. It aims to employ long- and short-range spatial interactions in the attention mechanism, leveraging the self-attention module. Organ-DETR also introduces DQM, an approach for one-to-many matching that tackles the difficulties in detecting organs. DQM increases positive queries for enhancing both recall scores and training efficiency without the need for additional learnable parameters. Extensive results on five 3D Computed Tomography (CT) datasets indicate that the proposed Organ-DETR outperforms comparable techniques by achieving a remarkable improvement of +10.6 mAP COCO and +10.2 mAR COCO. Code and pre-trained models are available at `https://---`.

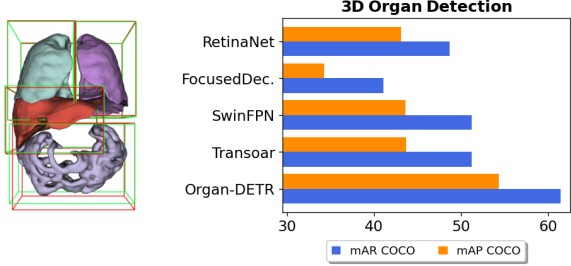

## 1 INTRODUCTION

The advent of the DEtection TRansformer (DETR) (Carion et al., 2020) represents a pivotal milestone in computer vision, specifically within object detection. DETR is distinguished by its novel query-embedding-based Transformer architecture that replaces traditional heuristic target box-to-anchor assignments (Ren et al., 2015). DETR fundamentally shifts the paradigm by directly mapping queries into distinct object representations. DETR-like methods have gone beyond object detection and have broadly impacted various vision recognition tasks like object detection (Zhang et al., 2022; Chen et al., 2022; Jia et al., 2023), segmentation (Dong et al., 2021; Cheng et al., 2021; 2022; Li et al., 2023b), pose estimation (Hampali et al., 2022), multi-object tracking (Meinhardt et al., 2022), and other pertinent domains.

The identification and delineation of organs hold significant implications in clinical applications, such as efficient data retrieval (Xu et al., 2019), robust quantification (Tong et al., 2019), and enhancement for subsequent tasks like semantic segmentation (Navarro et al., 2022; Azad et al., 2023). However, the transition from 2D to 3D visual data poses the challenge of processing data

in higher dimensions. Furthermore, medical imaging modalities, particularly 3D Computed Tomography (CT), inherently exhibit complexities pertaining to organ structures. These complexities encompass issues such as proximity and overlap, fuzzy boundaries, inter-patient variability, and different anatomical structures with analogous intensity levels (see Supplementary Figure 7). These challenges pose obstacles to effectively extending DETR-like approaches to 3D organ detectors, limiting their full potential.

This study makes the following primary contributions. *i*) We introduce a MultiScale Attention (MSA) module, a pioneering top-down representation learning approach adept at capturing both long- and short-range information from multiscale features. *ii*) We propose a Dense Query Matching (DQM) method, a novel one-to-many matching approach to boost recall and training efficiency. MSA is an attention-based encoder that captures high-level global context and dependencies between visual patches across different scales by leveraging a long-range cross-attention mechanism (see Supplementary Figure 8). The abstract features in high-level feature maps, like organs or instances, can provide invaluable guidelines for representing features in lower levels, like edges. In our organ detection transformer (Organ-DETR), we use MSA as an encoder and combine it with the decoder from Deformable DETR (D-DETR) (Zhu et al., 2020b). Our second contribution is a novel query matching strategy for the label assignment. One-to-one techniques (Carion et al., 2020; Zhang et al., 2023c) often suffer from slow training convergence and lower recall due to an insufficient number of positive queries. One-to-many methods (Chen et al., 2022; Kim & Lee, 2020; Zhu et al., 2020a; Ge et al., 2021; Feng et al., 2021; Shuai et al., 2022) partially solve the aforementioned issues but still suffer from difficult optimization, duplicate-removal operation, and low precision. We present DQM, a novel one-to-many label assignment strategy that addresses these challenges. A hyperparameter matching ratio in DQM governs the selection of positive queries to boost recall scores and the learning pace during training. We then rectify the intricacy of multiple predicted labels and bounding boxes through a proposed multiscale segmentation framework for expediting optimization procedures while elevating the precision levels of the outcomes. By integrating these innovations, our Organ-DETR shows superior performance over comparative methods, achieving an outstanding improvement of +10.6 mAP COCO and +10.2 mAR COCO across five widely-used CT databases.

## 2 RELATED WORK

**Feature representation**: In conventional object/organ detection, convolutional neural networks (CNNs) or Swin Transformers are commonly employed for feature extraction as backbones (He et al., 2016; Lin et al., 2017; Liu et al., 2021; Wittmann et al., 2022) or feature representations as DETR-like methods' encoders (Zhu et al., 2020b; Liu et al., 2022a; Cao et al., 2022; Li et al., 2023b; Zhang et al., 2023a; Li et al., 2023a). Prominent examples of widely adopted CNN models are ResNet (He et al., 2016), FPN (Lin et al., 2017), and Retina U-Net (Jaeger et al., 2020; Baumgartner et al., 2021). Due to the shortcomings of the CNN models in capturing long-range information, recent studies tend to employ Transformers in their detection architectures (Liu et al., 2021; 2022b; Wittmann et al., 2022). As shown in the experiments, existing Transformer-based encoders or backbones have exhibited limitations in performance, unfortunately. This inclination primarily arises from the necessity of a sufficient patch representation to effectively capture long-range information within Transformer-based structures. Furthermore, DETR-like methods often opt for high-level features due to the computational complexity, resulting in reduced spatial resolution. This constraint hinders the effective utilization of the Transformer-based features within the scope of detection tasks. In response to these challenges, we proposed the MSA module with a dual attention mechanism, strategically capturing a wide spectrum of long- and short-range feature patterns within and between layers (Figure 8). Recent studies underscore a growing preference for employing the multi-scale concept in Transformer models. Notably, FasterViT ? and MAFormer (Wang et al., 2022b) stand out for adeptly blending local and global features within a scale level. HiViT ? eliminates local inter-unit operations and retains only global attention between tokens through a series of spatial merge operations and MLP layers. MERIT (Rahman & Marculescu, 2023) focuses on utilizing multi-stage prediction maps for loss aggregation. In contrast to these techniques, MSA employs inter-scale cross-attention that generates key token voxel patches from between scales for the queries in the given scale, aiming to effectively capture more abstract information existing in high-level feature maps.

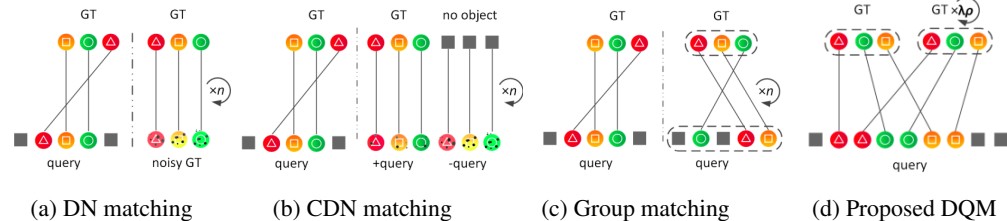

| (a) DN matching | (b) CDN matching | (c) Group matching | (d) Proposed DQM |

Figure 1: Overview of the different matching approaches: (a) DeNoising matching (DN) (Li et al., 2022); (b) Contrastive DeNoising (CDN) matching (Zhang et al., 2022); (c) Group matching (Jia et al., 2023); and (d) Proposed Dense Query Matching (DQM).

**Label assignemnt**: Conventional label assignment methods can broadly be classified into two categories: i) One-to-one strategy that assigns exactly one positive query for each ground truth instance, object, or organ (Carion et al., 2020; Zhang et al., 2023c); and ii) One-to-many strategy that allows the assignment of multiple positive queries to a single ground truth instance (Chen et al., 2022; Kim & Lee, 2020; Zhu et al., 2020a; Ge et al., 2021; Feng et al., 2021; Shuai et al., 2022). DETR (Carion et al., 2020) developed a one-to-one label assignment approach for directly transforming queries into distinct objects, thereby facilitating end-to-end object detection. The one-to-one label assignment has been improved by a considerable number of methods (Zhu et al., 2023; Zhang et al., 2023c). DeNoising (DN) methods (Li et al., 2022; 2023b) enhance the convergence of DETR by developing a query denoising scheme, where the ground truth labels and boxes are polluted by noises and then enforce the Transformer decoder to reconstruct ground truth objects given their noised versions (Figure 1a). Zhang et al. (2022) introduce a Contrastive DeNoising (CDN) training technique that enhances the DN by incorporating positive and negative samples of identical ground truths simultaneously (Figure 1b). DN and CDN enhance the robustness of predicted labels and bounding boxes, but their primary focus on denoising rather than detection yields lower recall.

One-to-many methods with dense query concepts are developed as an alternative to the one-to-one approach for incorporating more positive object queries for enhancing training convergence. Such techniques often increase positive samples and then post-process the predictions by non-maximum suppression (NMS) with a predetermined threshold (Chen et al., 2021; Wang et al., 2022a; Ouyang-Zhang et al., 2022). Recent one-to-many detection approaches refrain from using the NMS mechanism. This avoidance is rooted in NMS's tendency to introduce instability when eliminating duplicated predictions, leading to compromised recall and precision. Jia et al. (2023) poorpse Hybrid Matching (HM) that integrates the one-to-one matching branch with an auxiliary one-to-many matching branch during training without employing NMS. Group Matching (Chen et al., 2022) utilizes one-to-many to assign multiple positive object queries to each ground truth, then separates the queries into distinct, independent groups, and finally uses one-to-one to select the best positive query per object within each group (Figure 1c). However, these methods rely on one-to-one bipartite, making it challenging to achieve higher recall and maintain easy optimization during training, as previously discussed. The proposed DQM, as a one-to-many matcher, improves the true positive matching rate by increasing the ground truth instances, eschewing the need for auxiliary queries. The true positive rate is then controlled through the utilization of a hyperparameter matching ratio (represented by $\lambda$ in Figure 1d), strategically designed to enhance recall. Increasing the number of true positive instances in DQM simultaneously decreases the number of negative queries, leading to lower false positives (and thus, higher precision) and boosting the training's learning pace.

## 3    ORGAN-DETR

The principal stages for organ detection in 3D CT data are represented in Figure 2 and detailed in the following sections.

**Feature extraction backbone**: Considering visual CT data $I$ as a 3D array with dimensions $H \times W \times D$, where $H$, $W$, and $D$ represent the height, width, and depth, respectively. A CNN-based backbone, such as FPN (Lin et al., 2017; Wittmann et al., 2023) or ResNet (He et al., 2016), generates a set of $L$ feature maps. Within this set, $L_h$ high-level feature maps ($L_h \leq L$) are considered as input for MSA. Each feature map is then mapped into $f_e$ features by a convolutional layer, forming a feature representation $\{\mathbf{P}^{(l)}\}_{l=1}^{L_h}$, where $\mathbf{P}^{(1)}$ denotes the highest-level feature map.

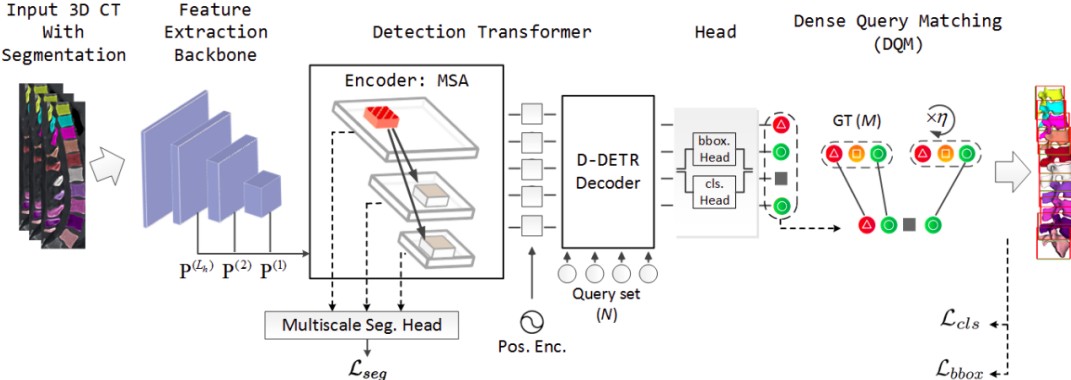

Figure 2: Organ-DETR overview: Extracted feature maps from input CT data by a backbone are fed into the encoder of Organ-DETR for being processed by MSA. MSA provides a deeper feature representation for the backbone's feature maps. A multiscale segmentation framework further guides the enriched feature maps before feeding them into the decoder, followed by the head. The query embeddings are finally matched through DQM, and the output is the prediction of class labels and bounding boxes of detected organs. Note organ segmentations are only available during training.

**Multiscale Attention**: Organ-DETR is a Detection Transformer that uses MSA as an encoder, and its decoder is borrowed from the Deformable DETR (Zhu et al., 2020b). The MSA encoder learns adaptive spatial sampling, allowing more flexible and accurate organ detection. As shown in Figure 2, the $L_h$ feature maps extracted by a backbone, $\{\mathbf{P}^{(l)}\}_{l=1}^{L_h}$, are fed into the MSA block. Global context information, like texture and patterns, tends to be repeated within a specific scale/layer or through different scales in feature and original data domains. By attending to distant patches, MSA can recognize complex patterns and objects that span a larger area of the visual data (see Figure 8 in Supplementary). This is particularly useful in the organ detection task, where understanding the relationships between distant regions is crucial for accurate predictions. Engaging distant patches across different layers in attention mechanisms also enables information flow across all patch embeddings, regardless of their spatial proximity. This helps propagate relevant information to all parts of the visual data, which is especially beneficial when dealing with large-scale visual data like CT, wherein organs are spread across different regions.

MSA is equipped with two attention mechanisms: i) *self-attention* that seeks a (short-range) attention map inside the given scale level, and ii) *cross-attention* that explores the attention map between interlayer patch embeddings, those located in other layers. We divide the backbone output feature maps $\mathbf{P}$ into $p \times p \times p$ non-overlapping voxel/3D patches, i.e. $\mathbf{X}^{(l)} = \{\mathbf{x}_1^{(l)}, \ldots, \mathbf{x}_{N_l}^{(l)} | \mathbf{x}^{(l)} \in \mathbb{R}^{p^3 \times f_e}\}$, $l \in \{1, \ldots, L_h\}$; $N_l$ denotes the number of voxel patches at the $l$-th scale level, and $\mathbf{x}_i^{(l)}$ is the vector representation of the $i$-th voxel patch at the $l$-th layer. The self-attention within the MSA is similar to traditional Swins' self-attention mechanism Liu et al. (2021) that computes the attention scores between all voxel patch embeddings in the given scale level for capturing short-range dependencies[1]. MSA projects $\mathbf{X}^{(l)} \in \mathbb{R}^{N_l \times p^3 \times f_e}$ into query, key and value using three matrices $\mathbf{W}_Q^{(l)} \in \mathbb{R}^{f_e \times f_q}$, $\mathbf{W}_K^{(l)} \in \mathbb{R}^{f_e \times f_k}$, and $\mathbf{W}_V^{(l)} \in \mathbb{R}^{f_e \times f_v}$, respectively:

$$\mathbf{Q}^{(l)} = \mathbf{X}^{(l)}\mathbf{W}_Q^{(l)}, \quad \mathbf{K}^{(l)} = \mathbf{X}^{(l)}\mathbf{W}_K^{(l)}, \quad \mathbf{V}^{(l)} = \mathbf{X}^{(l)}\mathbf{W}_V^{(l)}, \quad l = 1, \ldots, L_h. \quad (1)$$

Then self-attention $\mathbf{S}_{self}$ is computed via:

$$\mathbf{S}_{self}\left(\mathbf{X}^{(l)}\right) = \texttt{attention}\left(\mathbf{Q}^{(l)}, \mathbf{K}^{(l)}, \mathbf{V}^{(l)}\right) = \texttt{softmax}\left(\frac{\mathbf{Q}^{(l)}\mathbf{K}^{(l),\top}}{\sqrt{f_q}}\right)\mathbf{V}^{(l)}, \quad (2)$$

where '$\top$' denotes the transpose operation. The output of the self-attention block is fed into a hierarchy attention map, called cross-attention, that explores the attention scores between the voxel patch embeddings at layer $l$ and those located on its *higher* feature layers. Cross-attention is a

---

[1]Long-range refers to the voxel patch embeddings' relationship between different layers throughout this study.

long-range attention that explores the long-range relationship between inter-layer voxel patch embeddings. Since the natural visual data tends to repeat its features across several feature levels, it is tempting to capture such features to provide an impressive representation of the given scale level's patches. If $\mathbf{X}_0$ represents the output of the self-attention layer (that serves as the input for the computation of the cross-attention), the cross-attention layer $\mathbf{S}_{cross}^{(m)}$ is computed through:

$$
\begin{aligned}
\mathbf{S}_{cross}\left(\mathbf{X}_0, \mathbf{X}^{(m)}\right) &= \texttt{attention}\left(\mathbf{Q}_0, \mathbf{K}^{(m)}, \mathbf{V}^{(m)}\right) \\
&= \texttt{softmax}\left(\frac{\mathbf{Q}_0 \mathbf{K}^{(m),\top}}{\sqrt{f_q}}\right) \mathbf{V}^{(m)}, m \in \{1, \ldots, l-1\}.
\end{aligned}
\tag{3}
$$

Note that the highest-level feature map $\mathbf{P}_1$ contains no long-range cross-attention. Equation 3 is recursively applied across all $m$ values for yielding the long-range attention map for the given layer:

$$
\mathbf{X}_m = \prod_{m=1}^{l-1} \mathbf{S}_{cross}\left(\mathbf{X}_{m-1}, \mathbf{X}^{(m)}\right),
\tag{4}
$$

where $\mathbf{X}_0$ is the input feature map into the cross-attention module at the given layer $l$, i.e., $\mathbf{S}_{self}\left(\mathbf{X}^{(l)}\right)$. Following each multi-head self- and cross-attention layer, an MLP layer as a feed-forward network (FFN) is applied to voxel patch embeddings. The MLP comprises two linear transformations along with dropout layers, followed by ReLU non-linear activation functions. The Transformer encoder takes $L_h$ feature maps as input and produces output feature maps with identical resolutions. We guide the resultant feature maps of the encoder through a multiscale segmentation framework to yield relevant features (Figure 2). The segmentation head converts the $L_h$ refined feature maps into $L_h$ segmentation maps with $M$ labels and then are fed into the multiscale segmentation framework for computing the segmentation loss. In the Organ-DETR framework, we have adopted the decoder from Deformable DETR, which takes the enriched feature maps from the MSA encoder, query embeddings, and query position embeddings as its inputs and enhances feature representations to improve the accuracy of organ detection. A head predicts the bounding box coordinates and class labels of organs, and they are then fed into DQM for label assignment.

**Dense Query Matching**: DQM has been specifically designed for organ detection tasks, where each ground truth label corresponds to a single organ. Let $\mathbf{Q} \in \mathcal{R}^{N \times f_e}$, $\mathbf{q} \in \mathcal{R}^N$, and $\mathbf{g} \in \mathcal{R}^M$ represent query embeddings, the prediction scores of queries, and ground truth labels, respectively. A matcher assigns predictions to ground truth labels to form pairs for the detection task. Medical images are characterized by anatomical structures that have a similar appearance. For instance, the human vertebral column consists of tens of vertebrae that exhibit resemblant patterns. Owing to the resemblance observed among such structures, a potential approach entails elevating the recall rate and subsequently discerning the entities with the most favorable scores for designation as the ultimate predictions. Within this context, we propose DQM to enhance the relatively inadequate recall associated with the one-to-one assignment strategy while concurrently boosting learning during training. To improve the recall score, we add a new set of ground truth samples to the original ground truth during training (Figure 1d). We increase the number of positive queries to yield higher recall via duplicating the ground truth labels. The number of duplications is denoted by $\eta$ and defined as:

$$
\eta = \lceil \lambda \rho \rceil, \qquad \lambda \in (0, 0.9).
\tag{5}
$$

$\lceil . \rceil$ denotes a ceiling operator, and $\lambda$ is a hyperparameter, called *matching ratio*, which parameterizes a fraction of the potential queries that are matched to ground truth in a one-to-many setting; $\rho$ is the ratio between the number of queries and ground truth labels, i.e., $\rho = \frac{N}{M}$. DQM strictly assigns '$1 + \eta$' queries to each ground truth during training. In inference, DQM selects the best-predicted query per class/organ whose prediction score is the highest among the others. Hence, DQM generates only one prediction per ground truth, even though multiple queries are assigned to each ground truth. Hyperparameter $\lambda$ plays an important role in the matching performance as it determines the number of positive-negative queries and influences the pace of convergence. For a more in-depth analysis of the influence of the matching ratio $\lambda$, we analyze the loss function for a conventional one-to-one (O2O) approach. Considering the ratio of positive queries' gradient values between DQM and O2O, $\gamma^+$, we deduce in Appendix A that the following holds:

$$
\gamma^+ = \frac{\partial \mathcal{L}_{DM}}{\partial p^+} \Big/ \frac{\partial \mathcal{L}_{O2O}}{\partial p^+} = 1 + \lceil \lambda \rho \rceil.
\tag{6}
$$

Likewise, $\gamma^-$ denotes the ratio of negative queries' gradient values between DQM and the one-to-one mechanism, computed via

$$\gamma^- = \frac{\partial \mathcal{L}_{DM}}{\partial p^-} / \frac{\partial \mathcal{L}_{O2O}}{\partial p^-} = 1 - \frac{\lceil \lambda \rho \rceil}{\rho - 1}. \tag{7}$$

Figure 3 depicts the resulting equations for various values of $\lambda$ and $\rho$.

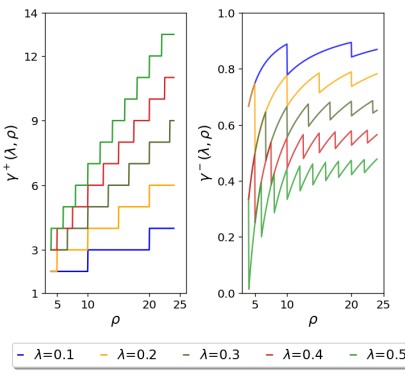

The graph demonstrates $\rho$ exhibits discrete incremental increments in $\gamma^+$, while decelerating $\gamma^-$. From an accuracy perspective, a larger number of queries in one-to-one methods is unfavorable as extra queries introduce confusion and complexity to the model, worsening the precision-recall scores. The figure further illustrates that an elevation in the matching rate $\lambda$ expedites the convergence of the gradient norm for positive queries while simultaneously decelerating the convergence for negative queries. Notably, acceleration is more prominent for positive queries than deceleration for negative ones. On the other hand, higher values for $\lambda$ increase the number of false positives, which can negatively impact the precision score. Therefore, the matching ratio $\lambda$ poses a trade-off between the precision-recall scores and the learning pace of the training phase. The ablation study in Section 4

Figure 3: Gradient ratio variation of positive (left) and negative (right) with different $\lambda$ values ($M$=10).

delves deeper into this aspect of DQM. We suppress the false positive rate by introducing a multiscale segmentation loss function detailed in the following section.

**Multiscale segmentation framework and training loss function**: Detection can benefit from segmentation. Recent studies (Li et al., 2023b; Wu et al., 2023) show that combining segmentation and detection tasks yields better performance compared to individualized approaches. Inspired by this, we introduce a multiscale segmentation loss function for maintaining the coherence between each layer of these feature maps and the corresponding input visual data. The encoder of Organ-DETR produces $L_h$ feature maps. Our segmentation framework employs a 3D convolutional layer to map $L_h$ feature maps into $L_h$ corresponding segmentation maps with the exact spatial resolution but comprising $M + 1$ channels, where $M$ is the number of ground truth labels and one denotes the background. The segmentation-related loss is computed using the cross-entropy (CE) and dice loss (DL) functions as follows: $\mathcal{L}_{seg} = \sum_{l=1}^{L_h} \left( CE^{(l)} + DL^{(l)} \right)$. As depicted in Figure 5, the multiscale segmentation framework incorporates the output of each layer of the MSA pyramid into the overall loss, enforcing the model to refine the predicted organs' labels and, consequently, their corresponding bounding boxes. The segmentation framework is exclusively utilized during the training phase to rectify the queries by suppressing the false negative queries and solidating the true positive ones. The final training loss function considers classification, bounding box, and segmentation aspects:

$$\mathcal{L} = \lambda_{cls}\mathcal{L}_{cls} + \lambda_{bbox}\mathcal{L}_{bbox} + \lambda_{seg}\mathcal{L}_{seg}. \tag{8}$$

$\lambda_{cls}$, $\lambda_{bbox}$, and $\lambda_{seg}$ correspond to the weights assigned to the classification, bounding box, and segmentation losses, respectively. The bounding box loss comprises both L1 and Generalised IoU (Rezatofighi et al., 2019), and the classification computes the cross-entropy loss between prediction and gound-truth logits.

## 4 EXPERIMENTS

**Datasets**: We applied the proposed Organ-DETR to five publicly available CT datasets including AbdomenCT-1K (Ma et al., 2022), WORD (Xiangde Luo & Zhang, 2022), TotalSegmentator (Wasserthal et al., 2022), AMOS (Ji et al., 2022), and VerSe (Sekuboyina et al., 2020; Löffler et al., 2020; Liebl et al., 2021) (Supplementary material Table 3). Among the datasets considered, all except VerSe comprise a mix of healthy and diseased organs, encompassing conditions such as cancer, tumors, and fatty liver. VerSe, as a vertebrae dataset, also includes instances of both healthy vertebrae and anatomically rare cases, such as fractured vertebrae and metal insertions. The

Table 1: Performance of different methods and ours averaged across five 3D CT datasets. RetinaNet is a CNN-based organ detector without a Transformer or matcher.

| Method | Backbone | Transformer | Matcher | mAP | mAR | $AP_{75}$ |
|--------|----------|-------------|---------|-----|-----|-----------|
| RetinaNet | FPN | - | - | 43.1 | 48.7 | 42.2 |
| - | UNETR+Swin | D-DETR | Hungarian | 40.4 | 47.6 | 35.0 |
| - | FPN | DETR | Hungarian | 32.7 | 39.6 | 26.6 |
| FocusedDec | FPN | Focused Decoder | Hungarian | 34.3 | 41.1 | 27.5 |
| SwinFPN | FPN + Swin | D-DETR | Hungarian | 43.6 | 51.2 | 38.4 |
| Transoar | FPN | D-DETR | Hungarian | 43.7 | 51.2 | 37.5 |
| Transoar | ResNet-50 | D-DETR | Hungarian | 41.4 | 48.6 | 35.1 |
| Organ-DETR | FPN | MSA + D-DETR | DQM | 54.3 | 61.4 | 56.0 |
| Organ-DETR | ResNet-50 | MSA + D-DETR | DQM | 51.2 | 57.9 | 50.2 |
| Organ-DETR | ResNet-101 | MSA + D-DETR | DQM | 51.5 | 58.1 | 48.9 |

Supplementary material Section B.3 reports additional information about datasets and preparation details.

**State-of-the-art organ detection techniques and evaluation metrics**: Our method's performance was compared against RetinaNet (Jaeger et al., 2020) as a CNN-based detector and several Transformer Detection methods including Focused Decoder (Wittmann et al., 2023), SwinFPN (Wittmann et al., 2022), and Transoar (Wittmann et al., 2022). RetinaNet is constructed based on the well-regarded medical object detection framework nnDetection (Baumgartner et al., 2021), incorporating minor modifications as outlined in (Wittmann et al., 2023). We also incorporate the pre-trained UN-ETR (Hatamizadeh et al., 2022) as a backbone in combination with D-DETR for additional analysis. To ensure a fair comparison among all detection methods, we maintain identical training settings, including the optimizer and its hyperparameters, batch size, and number of epochs. Parameters of similar blocks such as specifications of FPN and ResNets backbones (the number of scale levels, the number of feature maps employed in Transformer encoders, and the number of channels or features, etc.), Transformers' decoders (dropout, number of heads, embedding dimension, number of decoders, etc) remain identical across all methods, including ours. We used the recommended setting by the authors for other blocks like Swin Transformer, DETR, Focused Decoder, and Retina U-Net (Section B.5 in Supplementary). The methods' results were assessed by mAP COCO, mAR COCO, and $AP_{75}$, presented in Supplementary Section B.5.

**Organ-DETR parameters**: We used the following setting for the blocks dedicated to Organ-DETR. The MSA module was configured with a voxel patch size of 2, 64 attention heads, a depth of 2, and a dropout rate of 0.3, all without bias. In DQM, $\lambda$ and $\rho$ were configured as 0.2 and 10, respectively. During training, weights $\lambda_{cls}$, $\lambda_{bbox}$, and $\lambda_{seg}$ were specified as 2, 5, and 2, respectively.

**Main results**: Table 1 reports the performance averaged over the five CT datasets. The detailed results for each dataset are reported in Supplementary Section B.6.1. We depict the visual comparison of predicted bounding boxes by different techniques for the VerSe dataset in Figure 4. More results are provided in Figures 10-12 in the supplement. Organ-DETR demonstrates superior performance compared to all competing methods, showcasing substantial enhancements of +10.6 in mAP, +10.2 in mAR, and +13.8 in $AP_{75}$.

The superiority of Organ-DETR is consistent across all datasets, as evident from Tables 9 to 13. This underscores the method's reliability and effectiveness across diverse datasets with varying specifications. Table 1 also suggests that U-Net-based backbones like FPN may yield superior results to ResNet-based ones. The visual results confirm the quantitative results, where Organ-DETR bounding box predictions closely resemble reference annotations, outperforming other methods.

**Comparision of matching methods**: We compared the proposed DQM with Hungarian matching (Carion et al., 2020) and a range of recently developed matching methods including DeNoising matching (DN) (Li et al., 2022), Contrastive DeNoising (CDN) matching (Zhang et al., 2022), Hybrid Matching Jia et al. (2023), and matching with distinct queries Zhang et al. (2023b) for the organ detection task. All the matching methods, including DQM, were assessed under equitable conditions, utilizing the FPN Backbone and Deformable DETR. Unfortunately, 'memory mapping' and NMS in each *decoder* layer in the matching with distinct queries failed to work for the organ

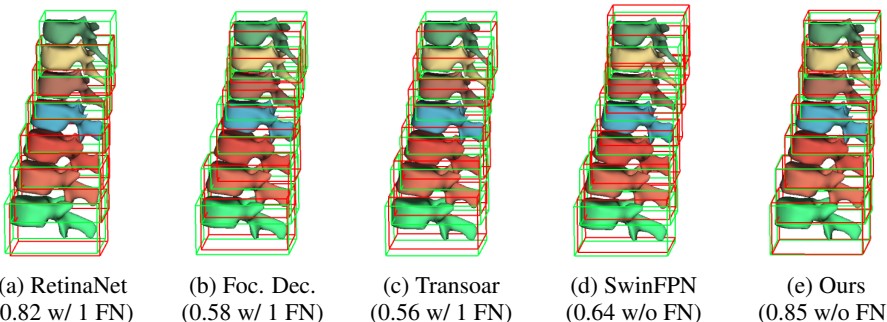



(a) RetinaNet | (b) Foc. Dec. | (c) Transoar | (d) SwinFPN | (e) Ours
(0.82 w/ 1 FN) | (0.58 w/ 1 FN) | (0.56 w/ 1 FN) | (0.64 w/o FN) | (0.85 w/o FN)



Figure 4: 2D visual comparison of bounding box and classification predictions (in red) by different organ detection methods within a sample from the VerSe dataset. The values in parentheses denote the average IoU over the predicated bounding boxes, and 'FN' indicates false negatives. Detected results are in Table 12 in the Supplementary.

Table 2: Comparison of the performance of different matching techniques in terms of AP COCO (AR COCO) averaged across five benchmark datasets.

| Method | Transformer | mAP | mAR | $AP_{75}$ |
|---|---|---|---|---|
| Hungarian | D-DETR | 43.2 | 50.6 | 37.3 |
| DN | D-DETR | 42.2 | 50.0 | 38.5 |
| CDN | D-DETR | 42.6 | 50.4 | 36.6 |
| Hybrid Matching | D-DETR | 43.8 | 51.3 | 38.5 |
| Matching with Distinct Queries | D-DETR | 36.9 | 43.7 | 30.2 |
| DQM (ours) | D-DETR | 44.5 | 52.0 | 42.8 |

detection task and were consequently omitted from the experiment. We employed NMS on the intermediate output of the Transformer encoder to obtain distinct queries, which are subsequently utilized within the Transformer decoder, so-called two-stage, in training and inference. Note that these techniques were not generically developed for organ detection tasks; hence, we found and reported the best setting for each matching technique throughout this study. For instance, the recommended setting for Hybrid Matching, $\{Q = 300, T = 1500, K = 6\}$, proved ineffective for organ detection, so we did an ablation study to find its best parameters for CT data which is $\{Q = 100, T = 300, K = 6\}$. Table 2 compares different matching strategies based on mAP, mAR, and $AP_{75}$, averaged across all five CT datasets. Detailed results with applied settings per dataset can be found in Supplementary, Tables 14-18. The tabulated results indicate that matching methods like DN, CDN, and matching with distinct queries are not recommended for 3D organ detection. In the context of CT data with a compact structure, as exemplified by the VerSe dataset (Table 15), introducing noise to the ground truth by DN and CDN is not advisable. Such noise addition can lead to ambiguity in the localization of closely situated organs, hence diminishing the suitability of DN and CDN for this specific data. NMS-based techniques like matching with distinct queries may be unsuitable for organ detection since NMS can exclude potentially valuable queries. The enhancement achieved by Hybrid Matching over the baseline one-to-one Hungarian method is marginal, whereas DQM significantly bolsters matching performance. This notable improvement is evident across most reported results in Tables 14 to 18, establishing DQM as a powerful tool for matching. In the supplement, Figures 13-14 provide a comparative analysis of sampling locations between the one-to-one approach and DQM across various samples during inference. Our DQM strategy incorporates more pertinent sampling locations in accurately predicting the bounding boxes.

**Impact of the multiscale segmentation framework on the accuracy of predicted classification labels**: Figure 5 demonstrates that the proposed multiscale segmentation framework boosts the matching performance. Since an increase in $\lambda$ may incur a higher number of false positive queries due to the increase of positive queries, the proposed segmentation framework rectifies the matched queries by reducing false positive queries. Hence, the introduced multiscale segmentation framework proves its capability to rectify class labels assigned to organs, leading to increased prediction accuracy, see Figure 5 right.

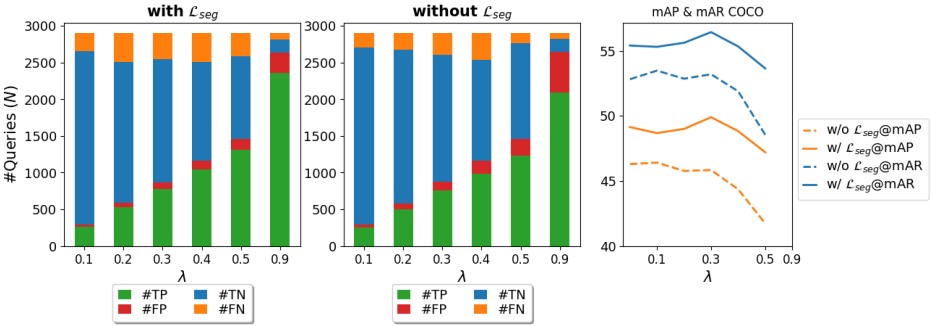

Figure 5: Performance of Organ-DETR (backbone: ResNet-50, $\rho = 10$) with and without the multiscale segmentation map on WORD in inference. T, F, P, and N stand for true, false, positive, and negative, respectively.

**Recommended $\rho$ and $\lambda$ values for organ detection**: Figure 6 displays the performance of Organ-DETR across a range of $\lambda$ and $\rho$ values in DQM. The values above the figure's main diagonal are more favorable, which indicates a higher averaged mAR-mAP and lower bounding box loss. Increasing $\lambda$ beyond a certain threshold while maintaining a constant $\rho$ generally results in an increase in bounding box loss. This is primarily due to a larger number of queries involved in the matching process. Likewise, increasing $\rho$ beyond a threshold while keeping $\lambda$ fixed can potentially worsen the performance. As a rule of thumb, $1 \leq \eta \leq 5$ yields desirable outcomes ($\eta = \lceil \lambda \rho \rceil$).

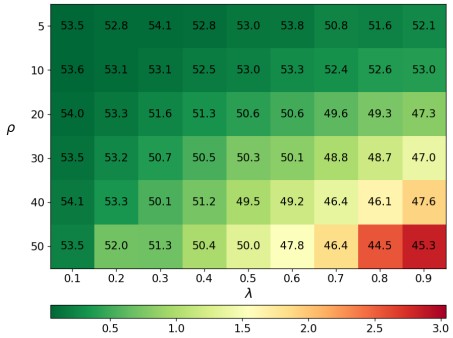

Figure 6: Ablation study on DQM parameters, i.e. $\rho$ and $\lambda$. The numerical values within the figure represent the average of mAP and mAR scores.

**Complement experimental results**: Due to the page limit, we reported the ablation study on MSA parameters in Supplementary Section B.6.1. We also report the results of DQM's gradient norm in the supplement, Section B.6.2. Last but not least, Supplementary Section B.6.6 reports computational cost, training time, and inference FLOPs.

## 5    DISCUSSION AND CONCLUSION

We introduced Organ-DETR for 3D organ detection, which includes two contributions: multiscale attention and dense query matching. The cross-attention of MSA improves different aspects of a backbone, including: *i) Global context and information flow*: Cross-attention allows complex pattern recognition and information flow across all scale levels in the visual data, regardless of their spatial proximity. Propagating relevant voxel patches to all parts of the data enables dealing with large visual data or scenes spread across different regions or a larger area of the visual data. *ii) Translation invariance*: Unlike traditional Transformers that rely on local patch embeddings, long-range patches in ross-attention allow to achieve a higher degree of translation invariance with a focus on important features and patterns in different layers of the visual data, irrespective of their absolute layer positions. *iii) Handling variable-sized visual data*: Thanks to the cross-attention mechanism, MSA can handle both high-resolution and low-resolution visual data without the need for resizing or cropping. The cross-attention ensures that relevant context is captured across various feature layers, regardless of the input visual data's dimensions. The impressive performance of Organ-DETR demonstrated that these contributions help to address the inherent challenges posed by 3D CT data, including proximity, overlap, and indistinct boundaries.

**Broader impacts**: Our results position Organ-DETR as a promising asset for future detection tasks in 3D imaging, shifting the attention toward 3D object detection.

REPRODUCIBILITY STATEMENT

To ensure the integrity and transparency of our research, we are committed to making our study fully reproducible. All reported experiments were performed on publicly accessible datasets. Supplementary Sections B.3 and B.4 provide comprehensive information on the datasets employed and the procedures involved in data preparation, preprocessing, and augmentation within the scope of this study. Experimental setups, hyperparameters, and training details are reported in Supplementary in Section B.5. We will release the codebase and the trained models publicly available on GitHub.

ETHICS STATEMENT

Our research paper does not raise ethical concerns.

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

# A APPENDIX: DQM'S GRADIENT NORM

Without loss of generality, the predicted labels can be categorized into binary classification, where queries possessing a positive probability score $p_i^+$ are regarded as foreground, and queries with a negative probability score $p_i^-$ are considered background. With a collection of $M$ ground truth instances and $N$ queries, the cross-entropy of the predicted labels can be defined as follows:

$$\mathcal{L}_{O2O} = -\sum_{m=1}^{M} log(p_m^+) - \sum_{n=1}^{N-M} log(1 - p_n^-). \tag{9}$$

Within the framework of the proposed DQM as a one-to-many approach, additional '$\eta M$' queries are identified as positive queries, while the remaining '$N - (1 + \eta)M$' candidates are classified as negative queries. The determination of $\eta$ is specified by equation 5, where it represents the count of replicated ground truth labels, a value that depends on the matching rate $\lambda$. Hence, the cross-entropy loss calculation for the DQM can be computed via:

$$\mathcal{L}_{DM} = -\sum_{m=1}^{(1+\eta)M} log(p_m^+) - \sum_{n=1}^{N-(1+\eta)M} log(1 - p_n^-). \tag{10}$$

By definition $\gamma^+$ as a ratio of gradient values between positive queries of DQM and baseline one-to-one, one can deduce the DQM's pace during treatment. Similarly, $\gamma^-$ is the ratio of gradient values between negative queries of DQM and one-to-one. They are calculated through:

$$\gamma^+ = \frac{\partial \mathcal{L}_{DM}}{\partial p^+} \Big/ \frac{\partial \mathcal{L}_{O2O}}{\partial p^+}, \quad \gamma^- = \frac{\partial \mathcal{L}_{DM}}{\partial p^-} \Big/ \frac{\partial \mathcal{L}_{O2O}}{\partial p^-}, \tag{11}$$

where

$$\frac{\partial \mathcal{L}_{O2O}}{\partial p^+} = \frac{\partial}{\partial p^+}\Big(-Mlog(p^+)\Big) = -\frac{M}{p^+} \tag{12}$$

$$\frac{\partial \mathcal{L}_{DM}}{\partial p^+} = \frac{\partial}{\partial p^+}\Big(-(1+\eta)Mlog(p^+)\Big) = -\frac{(1+\eta)M}{p^+} \tag{13}$$

$$\frac{\partial \mathcal{L}_{O2O}}{\partial p^-} = \frac{\partial}{\partial p^-}\Big(-(N-M)log(1-p^-)\Big) = \frac{N-M}{1-p^-} \tag{14}$$

$$\frac{\partial \mathcal{L}_{DM}}{\partial p^-} = \frac{\partial}{\partial p^-}\Big(-(N-(1+\eta)M)log(1-p^-)\Big) = \frac{N-(1+\eta)M}{1-p^-}. \tag{15}$$

By incorporating equations 12-15 into equation 11 and subsequently simplifying the expression, we arrive at the determined $\gamma^+$ and $\gamma^-$ ratios:

$$\gamma^+ = 1 + \eta = 1 + \lceil \lambda\rho \rceil, \tag{16}$$

$$\gamma^- = 1 - \frac{\eta}{\rho - 1} = 1 - \frac{\lceil \lambda\rho \rceil}{\rho - 1}. \tag{17}$$

