# OpenReview forum: "Organ-DETR: 3D Organ Detection Transfomer with Multiscale Attention and Dense Query Matching"
_ICLR.cc/2024/Conference — ICLR 2024 Conference Withdrawn Submission_

### Official Review · Reviewer_xGvm · 2023-10-17

**Soundness:** 4 excellent
**Presentation:** 4 excellent
**Contribution:** 3 good
**Rating:** 8
**Confidence:** 5

**Summary:**

This paper proposed a detection model called organ-DETR for 3D organ detection. There are two major designs, one for the multi-scale learning mechanism and another for the dense matching strategy. Extensive experiments on five public datasets demonstrate that the proposed organ-DETR has achieved superior performance than several public methods for 3D organ detection. The performance gain is impressive and the claims are supported by corresponding experiments. Overall, this work is sound.

**Strengths:**

1. Impressive performance gains
2. Extensive experiments
3. Clear motivation and reasonable model designs
4. Good paper writing

**Weaknesses:**

[1] This method is mainly designed and described from the technical view. However, for 3D organ detection, the clinical motivation and meanings are not clear. In other words, why are the ~10% improvements significant for medical tasks? The authors are suggested to elaborate more on this part.

[2] Top-down processing is a typical design for detection. The technical novelty should be clearly pointed out in this paper. Another issue is that ICLR might not be the best place for this paper and more technical contributions can be emphasized.

**Questions:**

See the above weaknesses.

---

> ### Author Response · Authors · 2023-11-14
> **Rebuttal Submission**
>
> We express our appreciation to the reviewer for providing the comments and suggestions. According to the comments, we plan to incorporate a paragraph to the Introduction that highlights the key contribution of this study. We also plan to edit the experimental section to highlight the improvements significant by Organ-DETR.
>
> We share the belief that this study is well-suited for ICLR, given its emphasis on novel methods in deep learning. Throughout the manuscript, we treat organ detection as a 3D object detection problem, emphasizing efficient representation for detection tasks and a reliable matching strategy.
>
> Once again, we appreciate all the comments, suggestions, and encouragement provided by the reviewer.  We hope we have addressed every comment satisfactorily. Your thoughtful input is greatly appreciated and we kindly invite the reviewer to raise any additional questions or points during the forthcoming reviewer-author discussion period, set to take place from Nov 10 to Nov 22.

---

### Official Review · Reviewer_nhJL · 2023-10-29

**Soundness:** 2 fair
**Presentation:** 2 fair
**Contribution:** 2 fair
**Rating:** 3
**Confidence:** 5

**Summary:**

This paper introduces Organ-DETR that leverages the power of Detection Transformers (DETR) for 3D organ detection in volumetric medical imaging. The authors identify the underexplored potential of applying query-based Transformers to 3D data and present two new modules, MultiScale Attention (MSA) and Dense Query Matching (DQM), to enhance the performance of Organ-DETR. The authors extensively evaluate Organ-DETR on five 3D Computed Tomography (CT) datasets. The experimental results demonstrate the superiority of the proposed approach over comparable techniques, showcasing a remarkable improvement of +10.6 mAP COCO and +10.2 mAR COCO.

**Strengths:**

- Adequate experiment and dataset: The authors conduct thorough evaluations on five 3D CT datasets, providing substantial evidence of the effectiveness of Organ-DETR.
- Extensive Evaluation: Author extensively analyze various parameters and conduct ablation studies, further enhancing the robustness of their experimental approach.

**Weaknesses:**

- Limited Novelty: The adoption of multi-scale attention in this paper is not novel, as it has been extensively explored in both medical imaging [1] and general visual recognition [2] contexts.
[1] Multi-scale Hierarchical Vision Transformer with Cascaded Attention Decoding for Medical Image Segmentation
[2] MAFormer: A Transformer Network with Multi-scale Attention Fusion for Visual Recognition
- One-to-Many Label Assignment: In the organ detection scenario, the use of a one-to-many label assignment strategy is not reasonable, considering that there is typically only one instance of each organ (e.g., liver, pancreas, spleen) in the human body.

**Questions:**

See weakness part.

---

> ### Author Response · Authors · 2023-11-14
> **Rebuttal Submission**
>
> We appreciate the comments and feedback provided by the reviewer. We would like to address the concerns raised regarding our method.
>
> Q1- Limited Novelty: The adoption of multi-scale attention in this paper is not novel, as it has been extensively explored in both medical imaging [1] and general visual recognition [2] contexts.
>
> A: The phrase "multi-scale attention" is broadly descriptive and caused confusion by suggesting that Organ-DETR is similar to earlier techniques such as those in references [1] and [2]. It is crucial to emphasize that Organ-DETR is quite distinct from these previous methods in terms of its methodology, conceptual framework, and practical implementation.
>
> In general, [1] and [2] use transformer-based encoders, while Organ-DETR uses a CNN-based encoder. MSA, as a module, refines the features for DETR. Both [1] and [2] lack cross-stage attention, while our MSA incorporates inner-stage self-attention and inter-stage cross-attention.
>
> In detail, MERIT [1] introduces a cascaded backbone with two hierarchical transformers, generating multiscale features that might appear similar to our method. However, the objectives for utilizing these multiscale features differ significantly. In MERIT, the multiscale features contribute to generating multi-stage prediction maps, employed in a multi-stage feature mixing loss aggregation to enhance segmentation. In contrast, our MSA module focuses on cross-attention between multiscale features to capture long-range feature patterns.
> In MAFormer[2], MAF blocks are employed within a single stage, lacking cross-stage operations. In contrast, our MSA module incorporates inner-stage self-attention and inter-stage cross-attention, effectively capturing long/short-range information. Additionally, the means to capture global features differ. MAFormer uses fully connected layers in the proposed Global Learning with Down-sampling (GLD) module, while our MSA module leverages cross-attention between multiscale features.
>
>
> Q2- One-to-Many Label Assignment: In the organ detection scenario, the use of a one-to-many label assignment strategy is not reasonable, considering that there is typically only one instance of each organ (e.g., liver, pancreas, spleen) in the human body.
>
> A: We have addressed this concern meticulously earlier in Sec. 2 and subsection “Dense Query Matching” in Sec 3. Below we briefly summarize DQM from the manuscript:
> The proposed DQM, as a one-to-many matcher, improves the TRUE POSITIVE MATCHING RATE by increasing the ground truth instances to enhance RECALL. Increasing the number of true positive instances in DQM simultaneously decreases the number of negative queries, leading to lower false positives (and thus, higher precision) and boosting the training’s learning pace. In INFERENCE, DQM selects the BEST-PREDICTED query per class/organ whose prediction score is the HIGHEST among the others. Figure 6 provides an ablation study on DQM parameters.
> Based on the comment, there may have been confusion by the reviewer about the exact strategy of DQM and, therefore, also the confusion about its mechanism. To fix that confusion, we plan to mention different steps of DQM in Sec. 3 clearly to make the DQM’s methodology easy to follow.
>
>
> Once again, we thank the reviewer for every comment. Our responses aim to address all concerns. If more questions or suggestions arise, we invite the reviewer to discuss them during the Nov 10-22 reviewer-author period. Your thoughtful input is highly valued, and we anticipate productive discussions during this time.

---

### Official Review · Reviewer_2UmR · 2023-10-29

**Soundness:** 2 fair
**Presentation:** 3 good
**Contribution:** 1 poor
**Rating:** 3
**Confidence:** 5

**Summary:**

The paper implements a Detection Transformer for medical organ segmentation. The approach claims some technical novelties for query matching, while the evaluation is restricted to finding organs that are very likely present and does exclude comparison to recent self-configuring detection frameworks (nnDetection).

**Strengths:**

The concept in itself is a reasonable approach to 3D multi-object detection. The paper comprises a reasonable number of ablations within the selected scope of methods. The approach is evaluated on real 3D CT data. The method is fast.

**Weaknesses:**

The experimental evaluation has in my opinion some important flaws and nowadays with the availability of fast and accurate 3D segmentation models it has limited clinically practical use. The authors only consider the detection of organs, which occur always exactly once in each scan. Object detection in medical 3D volumes when focussed on (healthy) organs is different from natural images, where the presence/absence of objects or multi-instances is the real challenge. The counterpart in medical imaging would be lesion/nodule detection, which may or may not appear once or multiple times in one scan. As a toy example the organ/bone detection task is fine, but a more realistic setting e.g. including LUNA16 as described in https://github.com/MIC-DKFZ/nnDetection would have to be considered. Table 1 only comprises adaptations of natural image detection pipelines to 3D but avoids direct comparison to realistic medical detection pipelines such as nnDetection or multi-label segmentation e.g. nnUNet. For VerSe (detection of vertebrae that could be well identified by a centre point) many more landmark localisation tools should have been evaluated in terms of localisation error in mm. Even within the comparisons RetinaNet show similar accuracy for VerSe despite slight differences in false positives. I wonder whether at least a direct comparison to the Retina-UNet which was part of the discontinued Medical Detection Toolkit (https://github.com/MIC-DKFZ/medicaldetectiontoolkit) could have been included?

**Questions:**

I would recommend to completely overhaul the experimental validation to include clinically more relevant medical detection tasks, e.g. LUNA16 and also incorporate SOTA methods. To my opinion this would require at least 50% change in content as the current submission is not directly evaluating multiple instances or localisation errors etc.

---

> ### Author Response · Authors · 2023-11-14
> **Rebuttal Submission**
>
> We appreciate the reviewer's comments and feedback on our study. We have taken into account all the comments and kindly request the reviewer to consider our responses, which we have addressed meticulously.
>
> Q1: The experimental has no clinically practical use…The counterpart in medical imaging would be lesion/nodule detection.
>
> A: As indicated by the title, our focus is on the detection of organs while lesions are not considered organs. Organ detection finds numerous applications for efficient data retrieval [1], robust quantification [2], improvement of downstream tasks (like semantic segmentation [3], etc), etc. We plan to allocate a section in the introduction to underscore the significance of organ detection in medical imaging.
>
> [1] Xu et al. "Efficient multiple organ localization in CT image using 3D region proposal network." IEEE transactions on medical imaging.
> [2] Tong et al. Disease quantification on PET/CT images without explicit object delineation, MedIA.
> [3] Navarro et al. "A Unified 3D Framework for Organs-at-Risk Localization and Segmentation for Radiation Therapy Planning, EMBC2022.
>
> Q2: Lack of comparison between medical detection pipelines such as nnDetection.
>
> A: nnDetection is embedded in the RetinaNet, we borrowed this technique from Focused Decoder [draft: Wittmann et al., 2023] and reported the results of this well-known technique in the paper! As the authors of Focused Decoder mentioned: “We adopt Retina U-Net from nnDetection with minor modifications..., we extract nnDetection's generated hyperparameters and refine them via a brief additional hyperparameter search…, converting it to a RetinaNet variant.” We apologize for the confusion and we would be happy to clarify it in the revised version. It is also incorrect that we did not compare our method with medical detection pipelines. FocusedDec, RetinaNet, SwinFPN, and Transoar, highlighted in the manuscripts, represent state-of-the-art and contemporary techniques for organ detection.
>
> Q3: RetinaNet performs similarly to our method on Verse despite slight differences in false positives.
>
> A: We believe the reviewer meant "False Negative" instead of "False Positive" and RetinaNet has one false negative, failing to predict the top vertebrae. Detailed results in Supplementary Table 12 show Organ-DETR outperforming RetinaUnet with a 55.1 mAP score compared to 46.3. While we can't include all results in the paper, we've attached visualization snippets in Supplementary Figs. 10-12. As stated, code and pre-trained models will be released for researchers to run and visualize results on their own.
>
>
> Q4: The authors only consider healthy organ detection.
>
> A: We trained and evaluated the methods on five different CT datasets that included healthy and unhealthy organs. Hence, it is not correct that we only consider healthy organ detection. As discussed previously, this study is devoted to organ detection and does not delve into topics such as lesion detection or distinguishing between healthy and unhealthy states. However, the insights provided in this study can serve as a beneficial reference for researchers in alternative detection domains seeking to enhance their approaches.
>
>
> Q5: For VerSe, many more landmark localization tools should have been evaluated in terms of localization error in mm.
>
> A: We explicitly explained the preparation process of datasets including Verse in Supplementary: Section B-3. Given that the primary emphasis of this study is on organ/object detection, the metrics employed—mAP, mAR, and AP75—are well-suited for detection tasks. While additional metrics could offer further assessment, our study emphasizes organ detection as a general task, demonstrating its applicability across various CT datasets with distinct functions. As mentioned earlier, we will release the code and pre-trained models, allowing readers to conduct additional measurements as needed.
>
> We trust that the provided responses address the reviewer’s concerns regarding Organ-DETR. We have taken diligent measures to ensure comprehensive coverage of Organ-DETR details within both the manuscript and Supplementary. As the reviewer-author discussion period is scheduled from Nov 10 to Nov 22, we eagerly await any forthcoming questions and will be delighted to offer further clarification during this time.

---

### Author Response · Authors · 2023-11-14
**Rebuttal Submission**

We sincerely thank the reviewers for their comments and feedback. We have incorporated their suggestions and responded comprehensively to their comments. Since the reviewer comments do not share common themes, we have addressed them individually. We hope that our study will be well-received as a valuable contribution to the ICLR’s focus on deep learning & application. We remain available for any further discussions or inquiries the reviewers may have.

Best regards,

Authors

---

### Meta-Review · Area_Chair_f8ur · 2023-12-06

**Metareview:**

This paper presents multiscale attention and dense query matching for 3D organ detection. Based on my reading of the paper, the comments, the response of the authors, I more agree with the reviewers that the multiscale attention is not novel. The authors shall either not claim this as novelty or improve their presentation to avoid the confusion (if any). A proper ablation study shall be provided to justify the benefits of two modules. Some visual results shall also be given to illustrate how the proposed modules improve the results qualitatively. The authors shall also pay more attention to the writing as there are many typos, e.g., many citations are not properly cited.

From the overall impact of this paper, I agree with the reviewer that "do not see the outstanding impact of the method in the scope a general machine learning conference". The authors may want to submit this work to medical imaging conference such as MICCAI.

**Justification For Why Not Higher Score:**

The work has several major weakness including novelty and experimental validation.

**Justification For Why Not Lower Score:**

NA

---

### Decision · Program_Chairs · 2024-01-16

Reject